# Heme enables proper positioning of Drosha and DGCR8 on primary microRNAs

Alexander C. Partin[1,2], Tri D. Ngo[1,2], Emily Herrell[1,2], Byung-Cheon Jeong[1,2], Gary Hon[1] & Yunsun Nam [1,2]

MicroRNAs regulate the expression of many proteins and require specific maturation steps. Primary microRNA transcripts (pri-miRs) are cleaved by Microprocessor, a complex containing the RNase Drosha and its partner protein, DGCR8. Although DGCR8 is known to bind heme, the molecular role of heme in pri-miR processing is unknown. Here we show that heme is critical for Microprocessor to process pri-miRs with high fidelity. Furthermore, the degree of inherent heme dependence varies for different pri-miRs. Heme-dependent pri-miRs fail to properly recruit Drosha, but heme-bound DGCR8 can correct erroneous binding events. Rather than changing the oligomerization state, heme induces a conformational change in DGCR8. Finally, we demonstrate that heme activates DGCR8 to recognize pri-miRs by specifically binding the terminal loop near the 3′ single-stranded segment.

---

[1] Cecil H. and Ida Green Center for Reproductive Biology Sciences and Division of Basic Reproductive Biology Research, Department of Obstetrics and Gynecology, University of Texas Southwestern Medical Center, Dallas, TX 75390, USA. [2] Department of Biophysics, University of Texas Southwestern Medical Center, Dallas, TX 75390, USA. Correspondence and requests for materials should be addressed to Y.N. (email: yunsun.nam@utsouthwestern.edu)

MicroRNAs (miRs) control the expression of most protein-coding genes in higher eukaryotes at the post-transcriptional level. The canonical miR maturation pathway requires the RNase III enzyme Drosha and its essential cofactor, DGCR8 (together forming the core Microprocessor (MP))[1–3]. Processing by Drosha requires a high degree of precision, because even a single-nucleotide deviation can affect which mRNAs are targeted[4–7]. Drosha activity is therefore highly regulated, and its dysfunction is frequently associated with disease[8,9]. Many mature miR levels do not correlate with the levels of primary microRNAs (pri-miRs)[10–13], underscoring the prevalence of post-transcriptional regulation of miR biogenesis.

A Drosha substrate—a pri-miR—can vary in structure and sequence but always includes a major stem-loop structure. At each end of the stem is a single-stranded/double-stranded RNA (ss–dsRNA) junction: the apical junction forming the terminal loop, and the basal junction at the opposite end. Drosha and DGCR8 have been suggested to cooperate to bind these two junctions, ensuring proper substrate recognition. Several primary sequence motifs have been identified in pri-miRs[14,15], which may help orient the complex, but these motifs are only present in a subset of pri-miRs, leaving hundreds of others apparently lacking recognition sequences. Therefore, in order to reliably bind and process pri-miRs, MP is likely to use the structural features of pri-miRs, but how such recognition is accomplished is unclear[16]. For many years, DGCR8 was associated with binding the basal junction, and the importance of the terminal loop was unclear due to conflicting results[17–19]. More recently, a crystal structure of Drosha has led to a revised model where Drosha binds to the basal junction, and recognizes a "UG" motif[20,21]. Assigning Drosha to the basal junction led to a model where DGCR8 recognizes a "UGU" motif in the terminal loop[20,22].

For more than a decade, DGCR8 has been known to interact with heme, via a heme-binding region (HBR or RNA-binding heme domain)[23]. Yet, how heme affects DGCR8 at the molecular level has remained unclear. Pri-miR processing is enhanced by heme both in vitro and in vivo, and variations of heme state (e.g., oxidation state and gas binding) can modulate DGCR8 activity, through unknown mechanisms[24–27]. Many heme-binding proteins utilize heme for chemical processes, such as catalysis, electron transfer, and gas transport[28]; however, heme can also serve as a signaling molecule, regulating diverse functions, such as transcription[29,30], ion flux[31], and cell signaling[32]. Thus heme is a versatile regulator of biological function and is expected to play a regulatory role in miR biogenesis. The HBR of DGCR8 does not share sequence homology with known domains, except for a WW motif (residues 298–352). Previous studies have proposed that heme serves to promote dimerization of DGCR8, where the thiol groups of Cys352 from two copies of DGCR8 serve as the axial ligands[23]. Dimerization has been suggested to help substrate recognition and accuracy[20,33]. However, how dimerization, heme binding, and pri-miR processing are mechanistically related is unknown.

Here we show that heme ensures fidelity of pri-miR processing. We show that heme is necessary for MP to cleave certain substrates at the correct site, and the degree of heme dependence varies for different pri-miRs. Heme dependence arises from failure of the basal junction to recruit Drosha; heme-bound

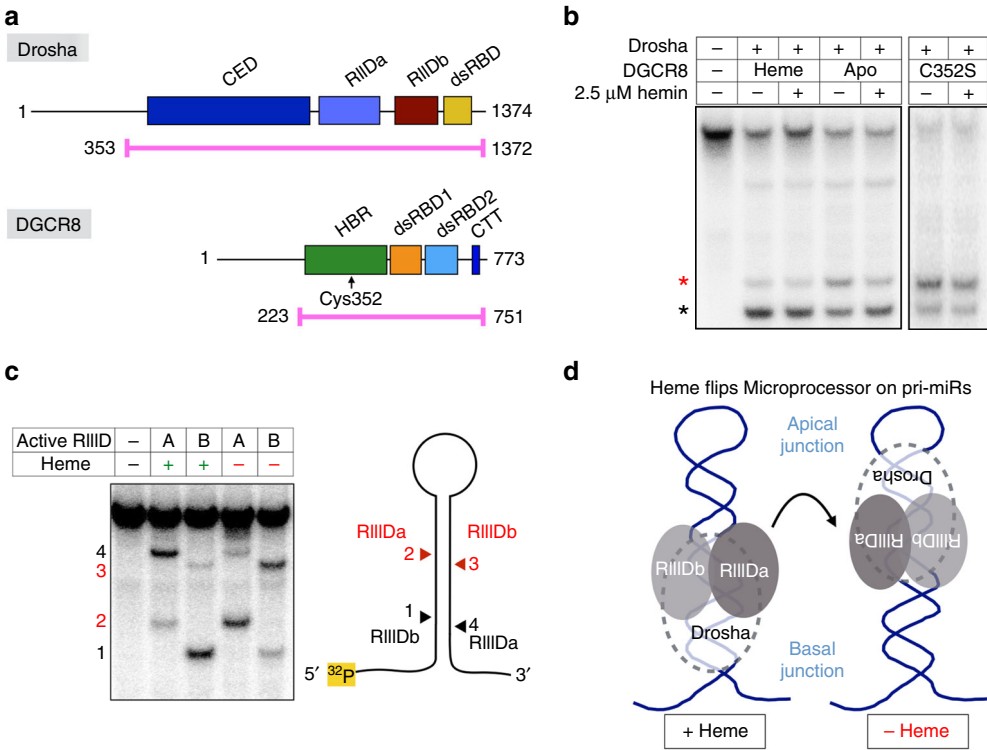

**Fig. 1** Heme binding reverses the orientation of Microprocessor on pri-miRs. **a** Domain organization and construct diagrams for Drosha and DGCR8. Recombinant protein complexes were purified after coexpressing the highlighted regions (magenta). Coexpression of Drosha with DGCR8[Heme], DGCR8[Apo], and DGCR8[C352S] yields MP[Heme], MP[Apo], and MP[C352S], respectively. CED central domain, RIIID ribonuclease III domain, dsRBD double-stranded RNA-binding domain, HBR heme-binding region, CTT C-terminal tail. **b** In vitro pri-miR processing assay using 5′ end-radiolabeled pri-miR-143. Reactions involving MP[Heme] (130 nM), MP[Apo] (130 nM) and MP[C352S] (260 nM) are shown. Correct product is shown with a black asterisk, incorrect product is shown with a red asterisk. **c** In vitro pri-miR processing assays used to orient the RIIID domains of Drosha on a pri-miR, with diagram shown to the right. Substrate shown is pri-miR-21. **d** Model depicting the heme-induced reversal of Drosha/DGCR8 complex

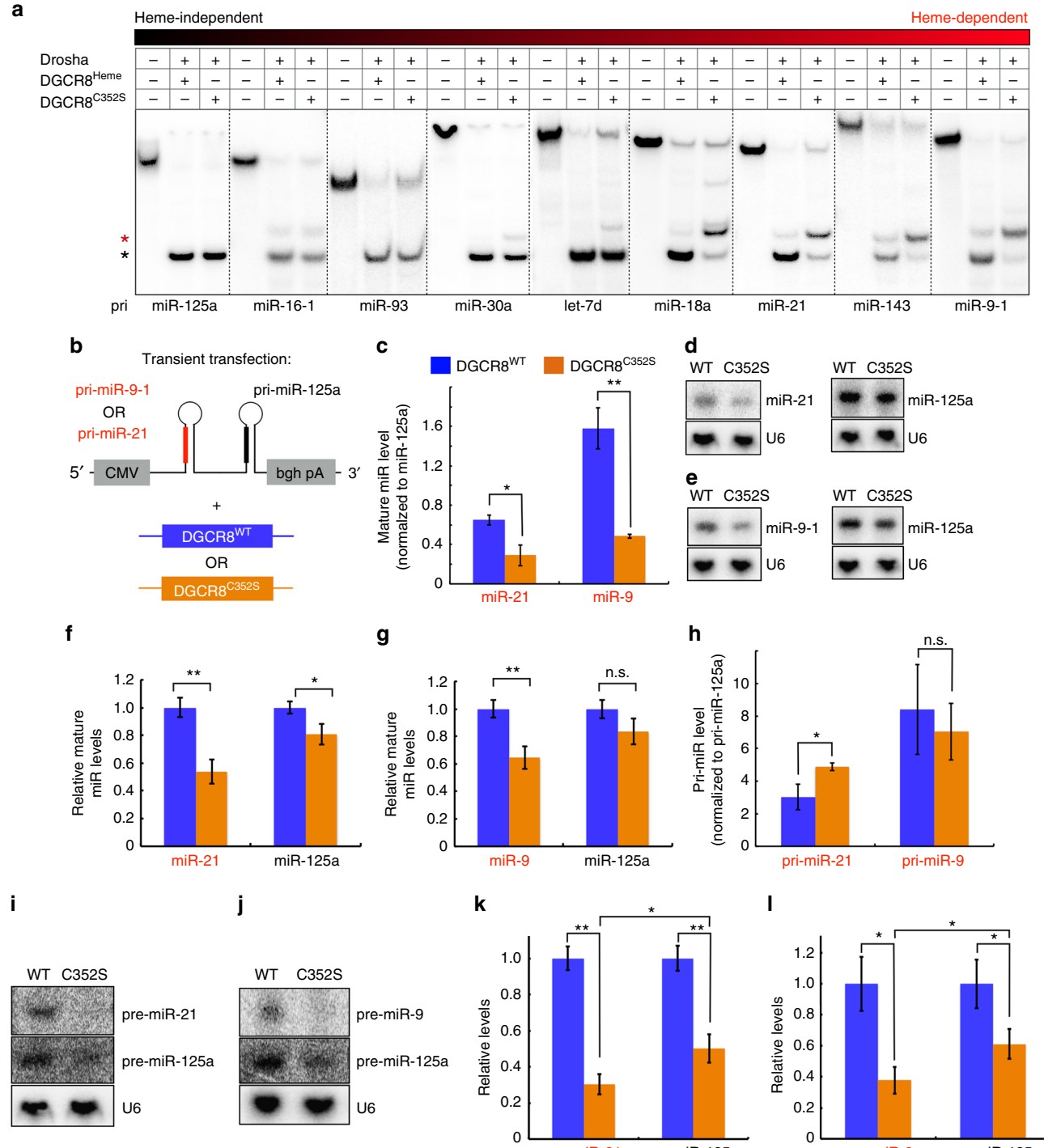

**Fig. 2** Heme dependence varies among miRs. **a** 5′ end-labeled in vitro processing assays comparing activities of MP$^{Heme}$ (130 nM) and MP$^{C352S}$ (130 nM) for a series of pri-miRs. Substrates are arranged from least heme dependent (left) to most heme dependent (right). **b** Cotransfection scheme for in vivo processing assays shown in **c–h**. Constructs containing pri-miR-9-1 and pri-miR-125a in tandem, or pri-miR-21 and pri-miR-125a in tandem, were cotransfected with either full-length FLAG-DGCR8$^{WT}$ or FLAG-DGCR8$^{C352S}$ into 293T cells. **c** Taqman quantitative PCR assay results from transfections in **b**. Mature miR levels were normalized to miR-125a expression levels. Data are represented as mean ± standard deviation from three biological replicates. Also see Supplementary Fig. 2a–d. **d**, **e** Representative Northern blots for pre-miRNAs indicated. U6 levels confirm normalized loading in each lane. **f**, **g** Quantitation of Northern blots as shown in **d** and **e**. Data are shown as mean ± standard deviation, for a total of three biological replicates. **h** Quantitative PCR of primary transcript levels for transfections in **b**. Levels were normalized to pri-miR-125a levels. **i**, **j** Representative splinted ligation assay results detecting miR-125a and miR-21 **i** or miR-9-1 **j**. U6 levels from Northern blots were used for normalization. **k**, **l** Statistical analyses of splinted ligation results, where miR levels with mutant DGCR8 were normalized to those with wild-type DGCR8. Data are represented as mean ± standard deviation, for a total of three biological replicates. *$p < 0.05$, **$p < 0.005$, n.s. (not significant), $p > 0.05$, Student's $t$-test (two-sided, paired)

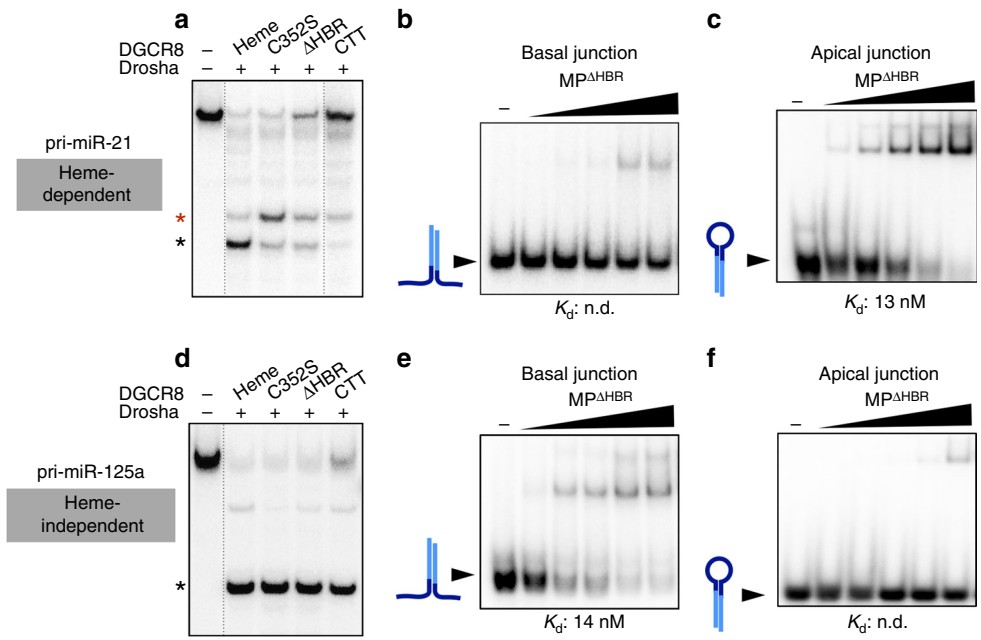

**Fig. 3** Heme enables DGCR8 to guide Drosha to the correct junction. **a** 5′ end-labeled in vitro processing assays of pri-miR-21, using MP$^{Heme}$ (130 nM), MP$^{C352S}$ (130 nM), MP$^{ΔHBR}$ (260 nM), and MP$^{CTT}$ (1.04 μM). Correct (black asterisk) and incorrect (red asterisk) products are indicated. DGCR8$^{ΔHBR}$ lacks the HBR but contains the tandem dsRBDs and C-terminal tail. Deletion of HBR and dsRBDs yields the C-terminal tail (DGCR8$^{CTT}$). **b, c** EMSAs comparing binding affinities of MP$^{ΔHBR}$ to basal **b** and apical **c** junctions of pri-miR-21. Schematic diagrams of the RNA are shown next to free RNA bands (arrow), with mature sequence shown in light blue. The basal junction RNAs are annealed RNA duplexes lacking the terminal loop. The apical junction RNAs are the pre-miR fragments. Protein concentrations are (left to right): 3.33, 6.67, 10, 13.32, and 16.67 nM. **d** 5′ end-labeled in vitro processing assays of pri-miR-125a, similar to **a**. **e, f** EMSAs similar to **b** and **c**, for pri-miR-125a

DGCR8 provides the necessary specificity to direct Drosha to the correct binding site, the basal junction. We show that DGCR8 is dimeric independent of heme binding, but dimerization alone is insufficient for accurate processing. Furthermore, heme induces a conformational change that activates the HBR to recognize the substrate RNA. We reveal that heme activates DGCR8 to recognize the structural context of the terminal loop of pri-miRs, and the binding event induces a unique RNA conformation near its 3′ end.

## Results

**Heme binding reverses MP orientation on pri-miRs.** In order to investigate the role of heme in miR biogenesis, we purified recombinant human Drosha and DGCR8 polypeptides from insect cells and isolated heme-saturated (MP$^{Heme}$) and heme-deficient (MP$^{Apo}$) species (Fig. 1a and Supplementary Fig. 1a). Our expression constructs contain all of the folded domains known to be required to cleave pri-miRs into canonical pre-miRs. We also purified the same complex with mutant DGCR8 (C352S)[23] that does not bind heme (MP$^{C352S}$). When Drosha/DGCR8 complexes are deficient in heme content, we observe a dramatic change in how they process pri-miR-143. In contrast to MP$^{Heme}$, both MP$^{Apo}$ and MP$^{C352S}$ generate a distinct, secondary cleavage product (Fig. 1b; red asterisk). When the MP$^{Apo}$ reaction is supplemented with 2.5 μM hemin, the proper cut site is reinstated. Adding heme to MP$^{C352S}$ does not change processing patterns, suggesting that the corrective effect of heme requires a specific coordination via Cys-352. Using sequencing gels and 5′ or 3′ end-labeled substrates, we identified that the wrong cleavage site lies in the middle of the mature miR sequence (Supplementary Fig. 1b, c). By combining the heme-binding mutation (DGCR8$^{C352S}$) with point mutations of the catalytic residues of Drosha, we determined the orientation of the MP complex in the absence of heme (Fig. 1c, d). The alternative cleavage event

associated with heme loss is caused by a "flipping" of the complex. Though reminiscent of "abortive processing" described previously for mutant substrates or domain deletions[17,20], what is striking is that wild-type Drosha and DGCR8 reverse their orientation on a wild-type pri-miR, depending on the mere presence of the heme molecule.

**Heme dependence varies among miRs.** We investigated whether alternative processing in the absence of heme occurs for other miRs. Using MP$^{C352S}$, we screened a series of pri-miRs for any changes in processing associated with heme loss (Fig. 2a). The extent of processing defect with the heme-binding mutant varies for different pri-miRs. Some pri-miRs (e.g., 9-1 and 21) are highly dependent on heme for processing at the correct site, while other pri-miRs (e.g., 125a) show little to no difference in processing patterns in the absence of heme. These results suggest that a subset of miRs, which we refer to as "heme-dependent," need heme for high-fidelity processing, whereas other miRs ("heme-independent") can be processed primarily at the correct site regardless of heme presence.

To assess how individual pri-miRs respond differently to heme in intact cells, we generated tandem pri-miR constructs for the expression in mammalian cells (Fig. 2b). A heme-independent pri-miR (125a) and a heme-dependent pri-miR (9-1 or 21) were expressed as part of a single transcript, along with either wild-type DGCR8 or the heme-binding mutant (DGCR8$^{C352S}$), in 293T cells. We quantified the amounts of intracellular mature miR levels by using both quantitative PCR (Fig. 2c and Supplementary Fig. 2a–d) and direct labeling via splinted ligation (Fig. 2d–g). In both assays, the DGCR8 mutation (C352S) is more detrimental to the production of mature miR-9 and miR-21 in comparison to the heme-independent control, miR-125a. Because the primary transcript levels of heme-dependent miRs are not lower in the cells with DGCR8$^{C352S}$, the heme-binding mutation

**Table 1 Results of SV-AUC and SEC-MALS experiments**

| | Predicted mass for monomeric DGCR8 (kDa) | Predicted mass for dimeric DGCR8 (kDa) | svAUC | | | | SEC-MALS | | DGCR8 state |
|---|---|---|---|---|---|---|---|---|---|
| | | | Sed. coeff. (S) | Molar mass (kDa) | Frictional ratio | RMSD | Molar mass by RI (kDa) | Zimm $R^2$ | |
| DGCR8$^{Heme}$ | 61.1 | 121.5 | 4.39 | 119 | 1.73 | 0.0049 | ND | ND | Dimer |
| DGCR8$^{C352S}$ | 59.8 | 120.9 | 4.05 | 120 | 1.94 | 0.0069 | 121.6 | 0.9894 | Dimer |
| DGCR8$^{Apo}$ | 60.5 | 120.9 | ND | ND | ND | ND | 125.3 | 0.9995 | Dimer |
| MP$^{Heme}$ | 179.4 | 240.5 | 7.50 | 230 | 1.59 | 0.0054 | ND | ND | Dimer |
| MP$^{C352S}$ | 179.4 | 239.8 | 7.31 | 215 | 1.57 | 0.0053 | 252.7 | 0.9962 | Dimer |
| MP$^{\Delta HBR}$ | 149.2 | 178.5 | ND | ND | ND | ND | 151.5 | 0.9125 | Monomer |

is likely to affect processing of pri-miRs after transcription (Fig. 2h). To confirm that the mutation affects production of pre-miRs, we also quantified the pre-miR levels using Northern blotting (Fig. 2i–l). Similar to mature miR levels, we observe less pre-miRs in cells expressing mutant DGCR8 and that heme-dependent miRs are more sensitive to the loss of heme compared to the heme-independent miR. Therefore, heme dependence varies among different miRs both in vitro and in cells and is an intrinsic property encoded into each pri-miR sequence.

**Heme enables DGCR8 to guide Drosha**. We asked why heme-dependent pri-miRs are processed differently from heme-independent pri-miRs. One possibility is that heme-bound DGCR8 is required to guide Drosha to the correct binding site, while the heme-free DGCR8 fails to do so. Another possibility is that heme-free DGCR8 actively directs Drosha to an incorrect site to give rise to processing errors. To determine the intrinsic behavior of MP lacking the HBR, we tested how a series of DGCR8 truncations affect processing of the two classes of pri-miRs (Fig. 3a, d, and Supplementary Fig. 3). As long as heme-bound HBR is present, Drosha can process both classes of pri-miRs at the canonical sites. However, when the HBR is removed, the processing accuracy is lost for heme-dependent pri-miRs, similar to MP$^{C352S}$, indicating that C352S is a loss-of-function mutation. When all of the RNA-binding regions of DGCR8 are removed—leaving only the minimal 23-residue C-terminal tail required for Drosha solubility (MP$^{CTT}$)[20]—Drosha cuts pri-miR-21, a heme-dependent substrate, predominantly at the wrong site. In contrast, Drosha cleaves pri-miR-125a, a heme-independent substrate, at the correct location regardless of which domains of DGCR8 are present. Therefore, Drosha is inherently error-prone for heme-dependent pri-miRs, whereas it can robustly recognize heme-independent miRs without any RNA-binding activity from DGCR8.

To determine whether Drosha intrinsically prefers the wrong site in heme-dependent pri-miRs, we tested its in vitro binding affinities for different junctions. Electrophoretic mobility shift assays (EMSAs) reveal that MP$^{\Delta HBR}$ binds to the apical junction of heme-dependent pri-miR-21 more tightly than to the basal junction (Fig. 3b, c). In contrast, a heme-independent substrate, pri-miR-125a, has a basal junction containing a stronger MP$^{\Delta HBR}$ binding site compared to its apical junction (Fig. 3e, f). Thus our results reveal that junctions in wild-type pri-miRs vary in their abilities to recruit Drosha, leading to a "tug-of-war" between the apical and basal ends. For heme-dependent pri-miRs where Drosha is drawn to the incorrect (i.e., apical) junction, heme-bound DGCR8 intervenes to position Drosha at the correct site. For heme-independent pri-miRs where Drosha finds the correct junction independently, DGCR8$^{Heme}$ likely provides additional contact to reinforce the overall binding affinity but is not necessary for processing fidelity. Therefore, the heme dependence of cleavage accuracy arises from the inability of the basal junction to outcompete the apical junction in recruitment of Drosha.

**Heme binding induces a conformational change but not dimerization**. Heme binding has been associated with homotypic interactions of DGCR8, but whether heme induces dimerization is unclear[21,23]. Using sedimentation velocity analytical ultracentrifugation (svAUC) (Supplementary Fig. 4a, b) and size-exclusion chromatography with multiangle static light scattering experiments (Supplementary Fig. 4c), we determined that heme-free DGCR8 is a stable dimer, both in isolation and in complex with Drosha (Table 1 and Supplementary Fig. 3a). Our results disagree with the previous proposal that heme promotes dimerization[23], possibly because DGCR8 without heme is prone to degradation that might be interpreted as monomerization[24]. Furthermore, we found that Drosha with DGCR8 lacking the HBR (MP$^{\Delta HBR}$) is a 1:1 heterodimer, suggesting that the HBR is necessary for forming heterotrimeric complexes. Thus our biophysical studies show that DGCR8 dimerizes via the HBR, in isolation as well as in the presence of Drosha, regardless of heme presence. Combined with our results described above (Fig. 3), we show that dimerization of DGCR8 is insufficient to guide Drosha to the correct target but that heme is the key factor required for high fidelity processing.

Because heme binding affects DGCR8 function without changing its oligomerization state, we questioned whether it produces an allosteric effect as seen in many other heme-binding proteins. Limited proteolysis of DGCR8 shows that the presence of heme affects its digestion profile (Supplementary Fig. 4d), suggesting a conformational change. Additionally, elution profiles from size-exclusion chromatography and the frictional ratios measured through svAUC suggest that heme-bound DGCR8 has a more compact conformation compared to the heme-free state (Table 1). Using negative stain electron microscopy, we show that MP$^{Heme}$ exhibits distinct features compared to the heme-free MP$^{C352S}$ particles (Fig. 4a, b). As expected from a conformational change, we also observe a shift in the melting temperature of DGCR8 depending on heme occupancy (Fig. 4c). Collectively, our results indicate that heme alters DGCR8 conformation rather than stoichiometry. To detect the functional outcome of the heme-induced conformational change, we performed EMSAs to compare RNA-binding activities of DGCR8$^{Heme}$ and DGCR8$^{C352S}$ (Fig. 4d). Upon binding heme, DGCR8 acquires dramatically higher affinity for pre-miR-143. Therefore, our biophysical and biochemical results suggest that heme acts as a molecular switch to induce a conformational change in DGCR8, which in turn activates DGCR8 to recognize stem loops.

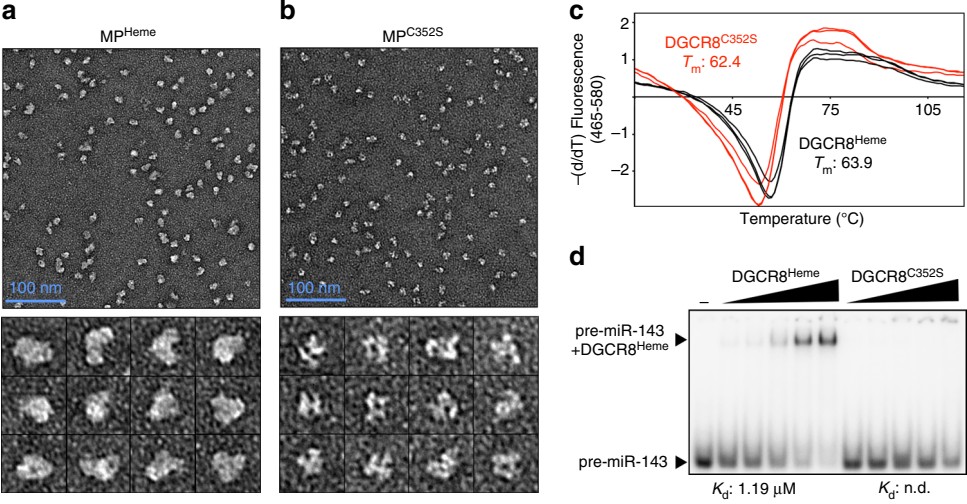

**Fig. 4** Heme induces a conformational switch in DGCR8 that facilitates RNA binding. **a**, **b** Representative negative stain electron micrographs of **a** MP$^{Heme}$ and **b** MP$^{C352S}$ after crosslinking with glutaraldehyde, with zoomed-in view of the selected particles shown underneath. **c** Thermal denaturation of DGCR8 in heme-saturated vs. heme-deficient states shows that the melting temperature ($T_m$) increases by 1.5 °C in the presence of heme. **d** EMSA with 5′ end-labeled pre-miR-143 shows that DGCR8$^{Heme}$ binds with an increased affinity compared to that of the DGCR8$^{C352S}$ mutant. "−" lane contains no protein. Protein concentrations are (left to right): 0.13, 0.26, 0.52, 1.04, and 2.08 µM

**Heme enables DGCR8 to recognize the terminal loop structure.** To dissect the molecular role of heme, we examined how heme affects specific protein/RNA interactions, using selective 2′-hydroxyl acylation analyzed by primer extension (SHAPE). SHAPE analysis on assembled pri-miR/Drosha/DGCR8 complexes reveals a striking difference between heme-bound and heme-free states. In the presence of heme, we observe a hyperreactive acylation site in the terminal loop of pri-let-7d, which would suggest a remodeling of the RNA backbone due to direct interactions with protein (Fig. 5a, b). This intense "hotspot" is absent for MP$^{C352S}$, even if the reaction is supplemented with extra heme (Supplementary Fig. 5a). The same nucleotide exhibits hyperreactivity when we add full-length, wild-type Drosha/DGCR8$^{Heme}$ complexes purified from 293T cells (Supplementary Fig. 5b). We also tested whether isolated DGCR8 is sufficient to recognize the terminal loop. Indeed, we observe the same SHAPE signature for DGCR8$^{Heme}$ but not for DGCR8$^{C352S}$, suggesting that DGCR8$^{Heme}$ specifically binds the terminal loop, independent of Drosha (Fig. 5c). To determine whether the specific interaction between the terminal loop and DGCR8$^{Heme}$ is conserved in other miRs, we tested for the presence of a SHAPE hotspot. We observe that many miRs exhibit intense levels of acylation at a confined site in the loop but only in the presence of MP$^{Heme}$ (Supplementary Fig. 5c–i). The SHAPE hotspots that we consistently observe are reminiscent of the hyperreactivity that accompanies unusual backbone geometry or base catalysis, which has been observed for certain stable protein/RNA complexes[34,35]. The conserved SHAPE hyperreactive site in terminal loops is thus likely due to a specific contact with the heme-bound DGCR8. Probing the hydroxyl radical reactivity of the loop region in the absence and presence of DGCR8 shows increased protection around the SHAPE hotspot (Supplementary Fig. 6a, b). Therefore, both SHAPE and hydroxyl radical footprinting data suggest that heme-bound DGCR8 directly binds the terminal loop.

The SHAPE hyperreactivity tends to focus around a single nucleotide, and this hotspot usually lies on the 3′ side of the ssRNA region in the terminal loop (schematics in Fig. 5a and Supplementary Fig. 5c–i)[36]. Remarkably, we observe these hotspots regardless of whether or not the pri-miR contains the apical UGU motif described previously[14,20], and the hotspots usually do not overlap with the UGU motifs. Aligning the terminal loop sequences according to the hotspot position reveals that many of them center around an adenosine (six out of the eight pri-miRs; Fig. 5d). To investigate the significance of the base identity, we examined how point mutations affect the in vitro binding affinity. In some cases, the base identity may play a modest role, as we observe reduced affinity when the hyperreactive adenosine of pri-let-7d is mutated to a cytosine (Supplementary Fig. 6c, d). However, in other miRs such as pri-miR-9-1, we observe no significant change in affinity for DGCR8 when we mutate the hotspot (Supplementary Fig. 6e, f), suggesting that sequence may not play a dominant role in affinity for certain miRs. As A49 in pri-let-7d makes a significant contribution to affinity, we tested how its base identity affects the SHAPE reactivity. Indeed, the same mutation that affects the affinity for DGCR8$^{Heme}$ also reduces the extent of hyperreactivity (Supplementary Fig. 6g).

Lack of sequence dependence in pri-miR-9-1 prompted us to test whether DGCR8$^{Heme}$ recognizes the structural context of the terminal loop. We investigated whether removing the constrained nature of the loop would eliminate the contribution of heme to DGCR8/miR interactions. A break in the terminal loop of pre-miR-9-1 reduces its in vitro affinity for DGCR8$^{Heme}$ to undetectable levels, similar to that of the heme-binding mutant, DGCR8$^{C352S}$ (Fig. 5e, f). Thus, heme confers upon DGCR8 the ability to detect the constricted structure of terminal loops in pri-miRNAs. Together, our data suggest that heme is necessary for DGCR8 to recognize the apical loop, through a combination of sequence and structural cues.

## Discussion

We present a model for how heme enables bipartite recognition of pri-miRs by Drosha and DGCR8 to ensure fidelity (Fig. 6). A pri-miR contains two ss–dsRNA junctions at each end of the stem, and Drosha must bind the basal junction to make the correct cut. The inherent structural symmetry near the branch points poses a unique challenge for MP to distinguish the basal junction from the apical junction. We reveal that heme-dependent miRs lack robust basal junctions to reliably recruit Drosha, resulting in Drosha reversing its orientation to bind the apical junction. In this tug of war between the junctions, heme

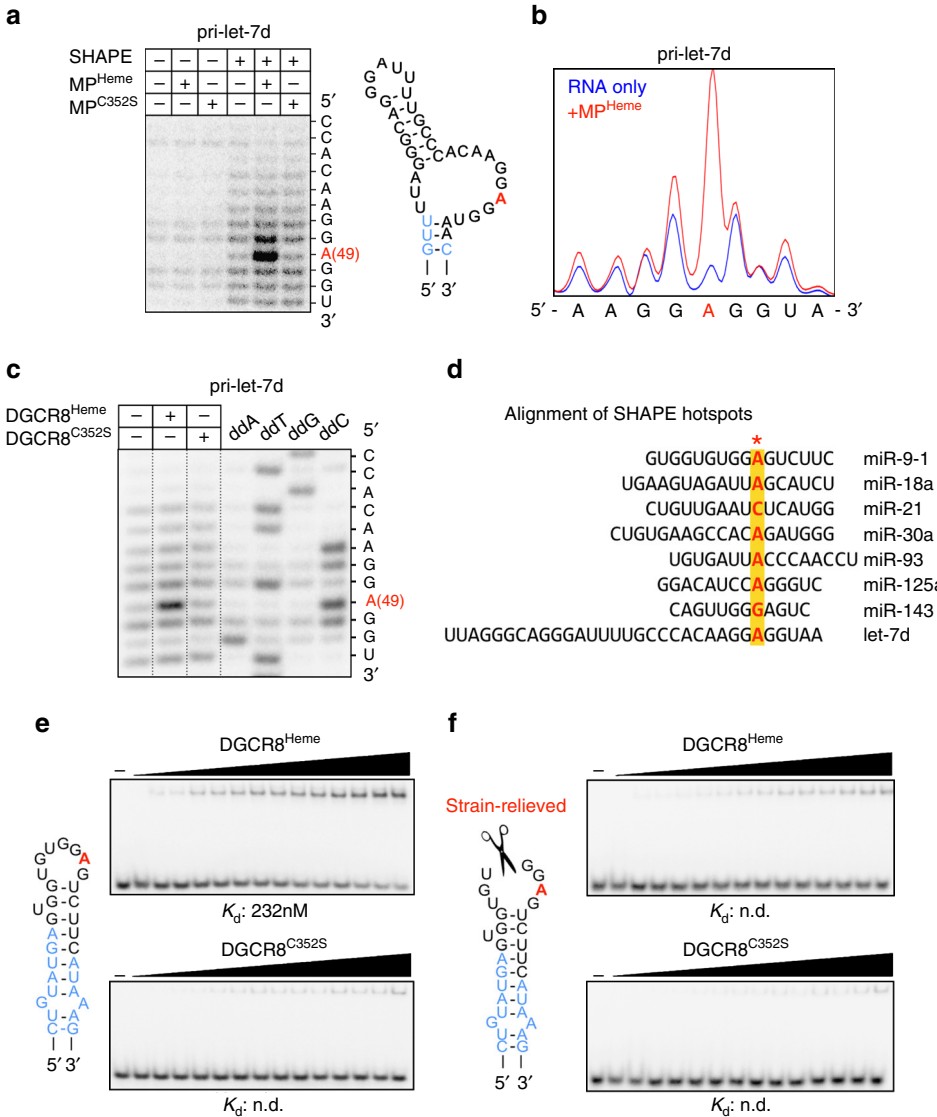

**Fig. 5** Heme binding allows DGCR8 to recognize the terminal loop structure. **a** SHAPE analysis of pri-let-7d, showing hyperreactivity associated with MP$^{Heme}$ binding. Secondary structure diagram (right) depicts the position of the SHAPE "hotspot" (red). Mature sequence is shown in light blue. Control reactions containing no SHAPE reagent (benzoyl cyanide) are shown in lanes 1–3. **b** SHAPE analysis of pri-let-7d with MP$^{Heme}$ by capillary electrophoresis. **c** SHAPE analysis of pri-let-7d with DGCR8. Only the reactions containing the SHAPE reagent are shown, and (unshifted) nucleotide marker lanes are shown to the right. **d** Diagram summarizing SHAPE results for a series of pri-miRs, aligned according to the site of highest reactivity marked with red font on yellow background. The sequence of the apical fragment released by Dicer is shown for each miR. SHAPE gels for additional miRs are shown in Supplementary Fig. 5c–i. **e**, **f** EMSAs comparing the binding affinities of DGCR8$^{Heme}$ and DGCR8$^{C352S}$ for **e** pre-miR-9-1 and **f** pre-miR-9-1 containing a central break in the terminal loop. Diagrams of RNA sequences are shown to the left. Protein concentrations are (left to right): 33, 50, 67, 83, 100, 117, 133, 150, 167, 183, 200, 217, 233, and 250 nM. All secondary structure diagrams were designed using mfold

tips the balance; heme confers upon DGCR8 the specificity for the terminal loop, thereby increasing the fidelity of the bipartite recognition.

What is notable is that a small (< 1 kDa) molecule—heme—is capable of globally altering the orientation of a large multiprotein complex (~ 300 kDa). For the previous decade, heme binding has been associated with dimerization of DGCR8 but its molecular role in pri-miR processing has been unclear. Here we show that heme does not change stoichiometry; rather, it converts a DGCR8 dimer into an active conformation capable of recognizing the terminal loop. Previous studies suggested that a UGU motif near the terminal loop acts as a binding site for the HBR, as mutating the UGU motif in miR-30a increases processing errors[20]. However, this experiment involved concomitant mutation of four nucleotides, one of which is immediately adjacent to the SHAPE

hotspot we identify in this study. Since only < 30% of pri-miRs contain the UGU motif, how the HBR interacts with the remaining pri-miRs is unknown. All pri-miRs we have tested exhibit a heme-dependent SHAPE hotspot, unless technical difficulties (such as reverse transcription artifacts) prevent detection. The unique SHAPE hyperreactivity caused by the HBR/terminal loop interaction identified in this study suggests a stable and specific protein/RNA complex with a distinct conformation. Using SHAPE, we uncover how heme enhances the RNA substrate specificity of DGCR8. Upon binding heme, DGCR8 gains the ability to recognize the constrained loop structure of the apical junctions of pri-miRs. In addition to the structural recognition, heme may also potentiate the ability of DGCR8 to have sequence preference near the SHAPE hotspot. As our results show, Drosha often fails to identify the correct ss–dsRNA

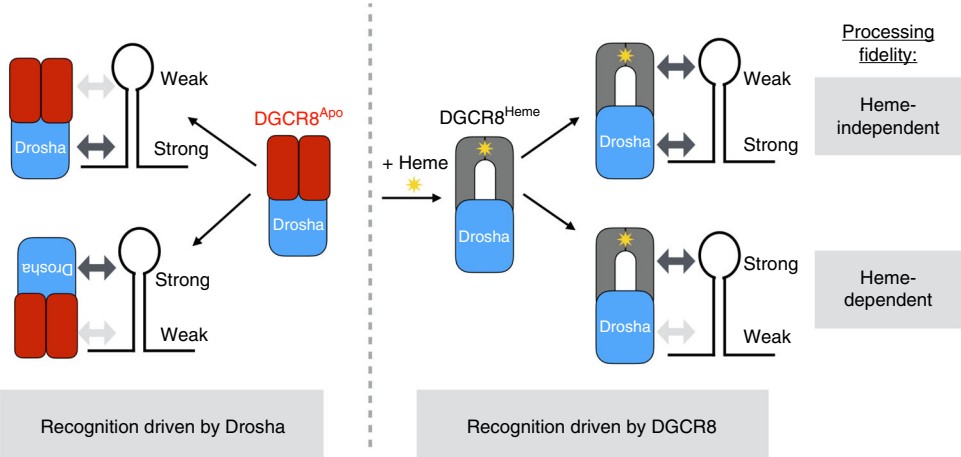

**Fig. 6** Model for heme-dependent pri-miR recognition by Drosha and DGCR8. Each pri-miR contains two ssRNA/dsRNA junctions where one is a stronger Drosha-binding site (indicated by "strong" or "weak"). Heme induces a conformational switch in DGCR8 that enables it to bind the terminal loop with higher specificity and affinity. Pri-miRs containing weak basal junctions are more heme dependent because placement of Drosha at the proper junction is driven by heme-bound DGCR8. For pri-miRs containing strong basal junctions, the interaction between the HBR and terminal loop can reinforce Drosha/pri-miR interactions to enhance processing efficiency

junction. However, heme allows DGCR8 to distinguish the apical loop from the basal junction, effectively enabling MP to break the symmetry of pri-miRs and support high-fidelity processing.

Our results suggest that heme dependence varies among different miRs. Heme-dependent pri-miRs contain stronger Drosha-binding sites in the apical junction rather than in the basal junction. Thus the extent of heme dependence is encoded into each pri-miR sequence. We reveal that not all wild-type basal junctions are ideal binding sites for Drosha, and recognition of pri-miRs with suboptimal basal junctions is especially dependent on heme-activated DGCR8. We propose that heme may serve as a signaling molecule to regulate miR biogenesis, and our in vitro studies and cell-based assays agree with this model. How heme affects the global miR expression profile in vivo and how other pathways might crosstalk with heme signaling to regulate miR biogenesis will need further investigation. For example, different heme states, such as changes in oxidation state or gas levels, could alter the relative levels of mature miRs. Our findings will aid such future studies to determine the physiological role of heme in regulation of miR biogenesis.

## Methods

**Protein expression and purification**. Drosha/DGCR8 complexes were expressed using the Bac-to-Bac system (ThermoFisher) in HighFive cells. Drosha and DGCR8 fragments were cloned into pFastbacDual, with a hexahistidine tag, and FLAG tag, respectively. The expression constructs contained the following residues: Drosha, 353–1372; DGCR8$^{Heme/Apo/C352S}$, 223–751; DGCR8$^{\Delta HBR}$, 489–751; and DGCR8$^{CTT}$, 728–751. Bacterial DGCR8 constructs in pET21 were expressed fused to hexahistidine tags in BL21 (DE3) Rosetta cells. For heme-saturated bacterial DGCR8, 5 µM hemin Cl (Sigma) was added to the media at the time of inoculation. Drosha/DGCR8 complexes and bacterial DGCR8 were purified by Ni-NTA affinity chromatography. Lysis and washing were performed using a buffer containing 20 mM Tris pH 8.0, 1 M sodium chloride, 10% glycerol, and 1 mM DTT. Cation exchange chromatography was performed using a 100–800 mM NaCl gradient buffered with 20 mM Bis-Tris pH 7.0, followed by a size-exclusion step. MP$^{Apo}$, the heme-deficient species, was separated away from MP$^{heme}$ through multiple rounds of ion-exchange chromatography.

**RNA transcription and purification**. RNA templates were cloned into pRZ vectors containing self-cleaving ribozymes on either end to produce homogenous ends (5′ hammerhead, 3′ hepatitis delta virus)[37]. Transcription reactions contained 1 µg linearized DNA template, 4 mM rNTPs, 35 mM MgCl$_2$, 50 mM Tris pH 8.0, 1 mM spermidine, 0.01% Triton-X 100, 10 mM DTT, and 100 U RNAse inhibitor (Thermo). Target RNA was purified by denaturing polyacrylamide gel electrophoresis (PAGE).

**In vitro pri-miR processing assays**. 5′ end-labeling was performed using T4 Polynucleotide Kinase (NEB) and γ-$^{32}$P-ATP. 3′ end labeling was performed using α-$^{32}$P-ATP and T4 RNA ligase 1 (NEB). Pri-miR processing assays were performed in 15 µL reactions containing 30 mM Tris pH 7.5, 67 mM sodium chloride, 5% glycerol, 10 mM MgCl$_2$, 5 mM DTT, 8 U RNase inhbitior, and 1.5 µg yeast tRNA. Reactions were incubated for 30 min at room temperature and analyzed by urea PAGE.

**Gel-shift assay (EMSA)**. 5′ end-labeled RNA was incubated with a dilution series of protein samples in 20 mM Tris pH 7.5, 140 mM sodium chloride, 10 mM DTT, 3 mM MgCl$_2$, 100 µg mL$^{-1}$ yeast tRNA, and 10% glycerol. The mixture was analyzed by native PAGE. For $K_d$ determination, the band intensities in each lane were quantified using the program ImageLab (BioRad), and fraction of RNA bound was plotted as a function of protein concentration. The data were fitted to a logarithmic curve and the concentration at which 50% of RNA was bound was reported as the $K_d$.

**Size-exclusion chromatography–multiangle light scattering**. Purified protein samples were loaded onto a Superdex 200 Increase 10/300 column (GE) using Agilent 1200 Infinity series HPLC, in a buffer containing 20 mM Tris pH 8.0, 750 mM sodium chloride, 1 mM TCEP, and 10% glycerol. DAWN HELIOS II and tREX detectors (Wyatt) were used to monitor scattering and differential refractometry. The results were processed using Astra 6.1 (Wyatt) and fitted using the Zimm model to obtain molar mass. For all samples, dn/dc was approximated to be 0.185 mL g$^{-1}$. Zimm $R^2$ values are shown in Table 1. Heme-saturated samples gave inconclusive results, likely due to interference from the heme signal, and are not reported.

**Sedimentation velocity analytical ultracentrifugation**. Samples were prepared at a series of concentrations between 0.1 and 1 mg mL$^{-1}$, in a buffer containing 20 mM Tris pH 8.0, 500 mM NaCl, and 1 mM TCEP and then loaded onto double-sector analytical cells. Using a ProteomeLab XL-I ultracentrifuge (Beckman Coulter), the cells were spun for approximately 16 h, at either RCF = 142,000×$g$ or 72,400×$g$, for DGCR8 and Drosha/DGCR8 complex, respectively, with ultraviolet (UV) and interferometry scans approximately every 10 min per cell. The data were date-corrected using the Redate program, then analyzed using SedFit[38] to obtain molar mass, sedimentation coefficient, and other parameters. For continuous c(S) distribution analysis, the simplex algorithm was used to fit the data, and the confidence level ($F$-ratio) was set to 0.68. The figures were created using the program GUSSI[39]. Root-mean-square deviation values are reported in Table 1.

**Electron microscopy**. MP$^{Heme}$ and MP$^{C352S}$ complexes were crosslinked using 0.1% glutaraldehyde and a protein concentration of 1 µM. The crosslinked samples were purified by size-exclusion chromatography. Negative stain grids were prepared using 2% uranyl acetate and the images were obtained using a Tecnai G2 Spirit TEM (FEI) running at 120 kV and using a magnification of ×30,000.

**Limited proteolysis**. α-Chymotrypsin (Hampton Research) was diluted to 0.01 mg mL$^{-1}$ in 10 mM HEPES pH 7.5 and 500 mM sodium chloride, then mixed 1:1 with DGCR8 protein (5 mg mL$^{-1}$) and incubated at 37 °C for 1 h. The reactions

were analyzed by sodium dodecyl sulfate (SDS)-PAGE and visualized by Stain-free (Biorad).

**Selective hydroxyl acylation followed by primer extension**. SHAPE reactions were performed on preassembled RNA-protein complexes. Protein and RNA samples were mixed at a final concentration of 312.5 and 65 nM, respectively, in 10 mM Tris pH 7.5, 130 mM sodium chloride, 50 μM $ZnCl_2$, 10 mM $MgCl_2$, 6.4 ng yeast tRNA, and 2 U Ribolock. SHAPE reactions were performed with 40 mM benzoyl cyanide (BzCN). Superscript III was used for reverse transcription using $^{32}P$-labeled primers and the resultant DNA was subjected to sequencing gel electrophoresis. To generate ladders, a separate reverse transcription was performed containing 1 mM of either dideoxy (dd)ATP, ddCTP, ddGTP, or ddTTP added alongside standard 0.5 mM dNTPs. These four ddNTPs were used to assign their respective Watson:Crick partners. Capillary electrophoresis samples were prepared with 6FAM and HEX-labeled primers and analyzed using a 3730XL Genetic Analyzer. The scans (.fsa files) were visualized using the QuShape program[40], and the traces were manually aligned according to the ddCTP ladder. After visualization of SHAPE results, the ladder must be shifted down by one nucleotide position, to account for the mechanism by which the SHAPE adduct interferes with reverse transcription. All secondary structure diagrams to locate SHAPE hotspots were generated by using mfold[36].

**Cell culture and quantitative real-time PCR**. 293T cells (authenticated and tested by ATCC) were maintained using Dulbecco's modified eagle medium supplemented with 10% fetal bovine serum and transient transfections were carried out using Lipofectamine3000 (Thermo) according to the manufacturer's instructions. Forty hours after transfection, total RNA and protein were extracted using Trizol (Thermo) and SuperScript II (Thermo) was used for reverse transcription. Mature microRNA levels were determined using the Taqman microRNA assay (Thermo), according to the manufacturer's instructions. Quantitative PCR was performed using the Taqman PCR Universal Master Mix II (no UNG) on a LightCyclerII (Roche). Primary transcripts and mature levels of miR-9-1 and miR-21 were normalized to pri-miR-125a or miR-125a by using comparative $C_T$ method. Statistical analyses were carried out for three biological replicates (three independent transfections) using paired Student's $t$-test (two tailed). Four technical replicates were performed for each value used in the analysis. For western blotting, protein samples from Trizol extraction were separated by SDS-PAGE and transferred to PVDF membranes. Anti-actin horseradish peroxidase (HRP; Sigma, A3854) dilution was 1:5000; Anti-FLAG M2 (Sigma, F1804) dilution was 1:1000. Goat anti-mouse HRP (Pierce, PI31430) dilution was 1:5000.

**Hydroxyl radical footprinting**. RNA–protein complexes were assembled as described in the SHAPE methods above, in a volume of 25 μL. To induce cleavage of RNA, a hydroxyl radical mixture (6 μL of 2% $H_2O_2$, 5.33 mM ammonium Fe(II) sulfate hexahydrate premixed with 50 mM EDTA, and 63 mM ascorbic acid) was rapidly assembled and mixed to the RNA–protein complex. After a 30-s incubation, the reactions were quenched, and reverse transcription was performed after acid phenol–chloroform extraction and ethanol precipitation, as described in the SHAPE methods above. The $^{32}P$-labeled DNA products were visualized by sequencing gel electrophoresis. For quantitation, band intensities were measured using ImageLab, and the differences in absolute intensities between buffer and protein lanes ($I_{buffer} - I_{protein}$) were reported for each nucleotide. The $y$-axis unit is defined as 25,000 volume units as measured by ImageLab. Nucleotides at positions 29 and 30 were excluded from the graph due to reverse transcription artifacts.

**Splinted ligation small RNA detection assays**. Splinted ligation reactions were performed as described previously[41]. Briefly, 3 μg of total RNA was annealed with 1 pmol of $^{32}P$-end-labeled ligation oligonucleotide and 1 pmol of a bridge nucleotide specific to the target miR, in a reaction containing 75 mM KCl and 20 mM Tris pH 7.5. The annealed sample was then incubated in a ligation reaction containing 30 mM KCl, 75 mM Tris pH 7.6, 10 mM $MgCl_2$, 1 mM DTT, 0.5 mM ATP, 7.5% w/v PEG 6000, and 2 U T4 DNA ligase (NEB). After a 1-hour incubation at room temperature, the reactions were stopped by adding 10 units of alkaline phosphatase and incubated at 37 °C for 15 min. The ligated samples separated by denaturing PAGE and visualized by autoradiography followed by quantitation using ImageLab. Statistics were carried out with three biological replicates for each group, using paired Student's $t$-test (two tailed).

**Northern blotting**. Total RNA (10 μg) mixed with equal volume of 2× formamide RNA loading dye was resolved using a denaturing polyacrylamide-urea gel. RNA was transferred to Hybond-N membrane in TBE buffer using Bio-Rad Trans-Blot cell followed by UV-crosslinking. Hybridization was performed in a buffer containing 0.5 M sodium phosphate pH 7.2, 7% SDS, and 1 mM EDTA with 10 ng of radiolabeled probe at 42 °C overnight. The membrane was washed in buffer containing 60 mM sodium phosphate, 1 M sodium chloride, and 6 mM EDTA at 42 °C for 30 min before exposing to a phosphor screen. The probe sequences for U6, pre-miR-125a, pre-miR-21, and pre-miR-9-1 are: GCAGGGGCCATGCTAATCTTC

TCTGTATCG, TCACAGGTTAAAGGGTCTCAGGGA, TCAACATCAGTC TGATAAGCTA and ACTTTCGGTTATCTAGCTTTAT, respectively.

**Data availability**. All detailed information and relevant data are available from the corresponding author upon request.

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

## Acknowledgements

We thank support from the Cecil H. and Ida Green Center Training Program in Reproductive Biology Sciences Research. We appreciate help from Chad Brautigam at the Macromolecular Biophysics Resource at UT Southwestern for AUC experiments. We thank the McDermott Center at UT Southwestern for assistance with capillary electrophoresis. We appreciate helpful comments on the manuscript from Kim Orth and Yuh Min Chook. Y.N. is a Southwestern Medical Foundation Scholar in Biomedical Research (Endowed Scholar Program at UT Southwestern), a Pew Scholar in the Biomedical Sciences, and a Packard Fellow (2013-39275). This study was supported by grants from the Welch Foundation (I-1851), Cancer Prevention Research Institute of Texas (R1221), American Cancer Society/the Harold C. Simmons Comprehensive Cancer Center (ACS-IRG-02-196), and NIH/NIGMS (5R01GM122960).

## Author contributions

A.C.P. and Y.N. designed the experiments and analyzed data. A.C.P. conducted the experiments. T.D.N., E.H. and B.-C.J. assisted with experiments. G.H. helped with cell-based experiments. A.C.P. and Y.N. wrote the paper.

## Additional information

**Competing interests:** The authors declare no competing financial interests.

