## [Peer Review File · Nature Communications]

Reviewers' Comments:

Reviewer #1:

Remarks to the Author:

The authors report nice findings to demonstrate the importance of heme binding for DGCR8 activity. The observation of heme-mediated DGCR8 binding and DROSHA correction are of high importance and provide novel insight into the mechanisms that regulated DGCR8 binding in a heme-dependent manner. The reports are very interesting and the biochemistry is nicely done. This is a nicely done manuscript that will be a good addition to the field!

I recommend for publication after the following experiments are completed or adequately addressed.

Figure 5 seems to suggest that the adenosine that is having higher SHAPE reactivity when bound is due to protein binding. I agree that SHAPE can be used to probe differences in flexibility, but the changes seem to be extremely focal, in comparison to those usually observed for RNA binding proteins. In binding most of them usually are around 10-12 nts in length for binding. In order to get a better handle on the structural change the authors should subject a limited number of RNA sequences to DMS profiling as well as hydroxyl radical probing. The DMS profiling would reveal structural changes for the adenosine of interest from changing the solvent accessibility of the W-C face of adenosine. This experiment would be advantageous as it would further our understanding of what is happening to the adenosine residue (as well as the other residues) that seems to be changing in many of the experiments. The hydroxyl radical probing would enable a more comprehensive analysis of protein binding, other than SHAPE. I stress this point because it is a major aspect of the paper -- the remodeling of structure due to DGCR8 binding.

The authors align a very small portion of the potenail pri-miRNAs (Figure 5, d) to see the adenosine hotspot. It would be worth analyzing and determining how many of the terminal loop sequences of all predicted pri-miRNAs have an adenosine within their terminal loop sequences. The modest alignment makes it seem as though their findings are very limited and ARE NOT a broad paradigm for pri-miRNA processing. This needs to be addressed.

Lastly, for their gel shifts, I did not find a true negative control for the loop structure. It is essential that the authors demonstrate that a random loop with an adenosine cannot bind to the proteins.

Reviewer #2:

Remarks to the Author:

Review of "Heme enables proper positioning of Drosha and DGCR8 on primary microRNAs"
manuscript

This is a highly interesting paper demonstrating that DGCR8 binding of Heme facilitates pri-miRNA processing by enforcing correct orientation of the microprocessor complex on pri-miRNA. Moreover, the authors show that only a subset of miRNAs require the heme binding to ensure correct processing. These are novel findings, which significantly expands the understanding of miRNA processing. The manuscript adds new knowledge on top of the already known dependency of the microprocessor complex on Heme and makes a strong case that Heme does not controls the dimerization of DGCR8, as previously reported. The presented experiments are well designed with good controls, the quality of the figures is high and the paper is well written.

The only experiment, which I think should be improved before publication is the demonstration of the mechanism being active in transfected cells, which is based on detection of the levels of mature miRNA expressed from a transfected construct (figure 2b). I find that it is important to show that the cleavage positions of the Heme dependent miRNAs in cells are indeed dependent on Heme-loaded DGCR8. Preferably, this could be by performing small-RNA sequencing on transfected cells. Alternatively, the faulty processing of Heme-dependent miRNAs in cells could be demonstrated by detecting the miRNAs and pre-miRNA with northern blots.

Major points:

1. In my opinion, the manuscript would be significantly improved if the in-cell validation experiment presented in figure 2b-d were supplemented with a sequencing based analysis of the small RNA content of the cell. This would make it possible to investigate many more miRNAs than the ones presented in the manuscript and could allow the aberrant cleavage events for the Heme dependent pri-miRNAs to be directly identified, which would demonstrate the in vivo relevance of the convincing in-vitro findings of the manuscript. Moreover, identification Heme dependent processing of endogenous pri-miRNAs is important to determine how wide-spread the phenomenon of Heme dependent processing is.
2. Statistical analysis of the data presented in figure 2b should be included.
3. In the closing remark of the manuscript the authors speculate that the Heme dependence of some miRNA may act as a means for regulation by changes in oxidation state or gas levels. This is indeed a highly interesting possibility, which would be relatively easy for the authors to investigate with their well established in vitro setup.

Minor comments:

1. The mechanism by which DGCR8 are able to detect the loop structure of the pri-MiRNAs is interesting. A more careful analysis of the loop requirements for DGCR8-Heme binding could possibly provide more insights. The SHAPE-probing clearly shows that the presence of DGCR8-Heme make specific positions in the pri-miRNA loops hyper reactive. The authors suggest that "remodeling of the RNA backbone due to direct interactions with protein" is causing the hyperreactivity. However, a larger remodeling of the loop would likely result in more positions changing reactivity than what is observed here. Hyper reactivity to SHAPE reagents is not observed that frequently and is typically caused either by the position being locked in a rare reactive backbone conformations or by base catalysis of the reaction (McGinnis et al, JACS 2012). I therefore think that it would be relevant to include some more discussion of how the Heme loaded DGCR8 could cause the hyperreactivity.
2. The mature miRNA 5' end is defined by either Drosha or Dicer. It would be interesting, if the Heme dependence of correct Drosha processing correlates with the need for accurate Drosha definition of the seed sequence of different miRNAs. Is this the case?
3. P8, line 15-16: please elaborate on how more degradation could lead to Heme dependent dimerization.
4. In figure 1: The use of green/red coloring to designate Heme vs. non-Heme is not optimal and makes it difficult for the red/green color insecure individuals to interpret the figure. In addition, panel b could go to supplementary, whereas it would make sense to show the data now presented in in supplementary figure 1c in the main figure, as it is the most direct evidence of the MP flipping around.
5. P7, line 2, p8 line 21: "correct Drosha" is used, but "guide Drosha to the correct binding site" or similar would be more precise.
6. Supplementary Fig. 5a should include the non-SHAPE probed controls and the loop basepairing

should only be indicated if the probing pattern support it. Information about the ladder used for these blots should be included.

7. The SHAPE probing pattern of the point mutants investigated in Supplementary Fig. 5j-m would be an interesting addition to the paper.

8. P12 line 16-18: this statement is too confident, given that only a single precursor was investigated.

9. P17 line 4: describe more carefully the method used for Q-PCR normalisation.

Responses to the reviewers' comments

We thank the reviewers for taking the time to review our paper. We appreciate that both of the reviewers share our enthusiasm for the work presented in the manuscript, as it elucidates a heme-dependent mechanism that helps us shape a molecular model for how Microprocessor works. We are also thankful for all the helpful suggestions, as following them has enabled us to improve our manuscript further.

Reviewers' comments:

Reviewer #1 (Remarks to the Author):

The authors report nice findings to demonstrate the importance of heme binding for DGCR8 activity. The observation of heme-mediated DGCR8 binding and DROSHA correction are of high importance and provide novel insight into the mechanisms that regulated DGCR8 binding in a heme-dependent manner. The reports are very interesting and the biochemistry is nicely done. This is a nicely done manuscript that will be a good addition to the field! I recommend for publication after the following experiments are completed or adequately addressed.

We appreciate the reviewer's positive comments for the findings we report in this manuscript. We agree with the reviewer that we are tackling an important question and that our findings provide significant progress.

Figure 5 seems to suggest that the adenosine that is having higher SHAPE reactivity when bound is due to protein binding. I agree that SHAPE can be used to probe differences in flexibility, but the changes seem to be extremely focal, in comparison to those usually observed for RNA binding proteins. In binding most of them usually are around 10-12 nts in length for binding. In order to get a better handle on the structural change the authors should subject a limited number of RNA sequences to DMS profiling as well as hydroxyl radical probing. The DMS profiling would reveal structural changes for the adenosine of interest from changing the solvent accessibility of the W-C face of adenosine. This experiment would be advantageous as it would further our understanding of what is happening to the adenosine residue (as well as the other residues) that seems to be changing in many of the experiments. The hydroxyl radical probing would enable a more comprehensive analysis of protein binding, other than SHAPE. I stress this point because it is a major aspect of the paper -- the remodeling of structure due to DGCR8 binding.

We agree with the reviewer that an independent method to analyze the RNA-protein interactions would be helpful. We now include hydroxyl radical probing data for the DGCR8:pri-miRNA complex in Supplementary Fig 6a-b. As the reviewer pointed out, a real footprint should span more than 1-2 nucleotides. The location of the footprint we identify is a larger area that surrounds the SHAPE hotspot, and thus agrees with our model that DGCR8^{heme} interacts with the terminal loop region including the hotspot. In the text we elaborate (end of p.12) that the SHAPE hotspot is indicative of a unique chemical environment for the particular nucleotide upon complex formation, which is caused by a direct interaction between DGCR8 and the RNA in the loop. Although we predicted this, we were able to provide better experimental evidence and describe the model better thanks to the reviewer's suggestion.

The authors align a very small portion of the potenail pri-miRNAs (Figure 5, d) to see the adenosine hotspot. It would be worth analyzing and determining how many of the terminal loop sequences of all predicted pri-miRNAs have an adenosine within their terminal loop sequences. The modest alignment makes it seem as though their findings are very limited and ARE NOT a broad paradigm for pri-miRNA processing. This needs to be addressed. Lastly, for their gel shifts, I did not find a true negative control for the loop structure. It is essential that the authors demonstrate that a random loop with an adenosine cannot bind to the proteins.

Although we often observe a hotspot at an adenosine nucleotide, our results (Supplementary Fig. 6c-f) suggest that the base identity only sometimes contributes to the binding affinity (by 2 fold at best, only for let-7d in our data). Thus, our model focuses on the structural features of the loop (which becomes more protected from hydroxyl radicals in the presence of heme-bound DGCR8) rather than the presence of an adenosine. Because every pri-miRNA has a terminal loop, as the reviewer suggests, our findings do indeed shape the general paradigm. The individual contributions of loop structure and base identity might be different for each microRNA, and such molecular details will require further studies (eg. structures of the complex). Prompted by the reviewer's comments, we have revised the text (p. 13 lines 13-15) to ensure that this is clear.

Reviewer #2 (Remarks to the Author):

Review of "Heme enables proper positioning of Drosha and DGCR8 on primary microRNAs" manuscript

This is a highly interesting paper demonstrating that DGCR8 binding of Heme facilitates pri-miRNA processing by enforcing correct orientation of the microprocessor complex on pri-miRNA. Moreover, the authors show that only a subset of miRNAs require the heme binding to ensure correct processing. These are novel findings, which significantly expands the understanding of miRNA processing. The manuscript adds new knowledge on top of the already known dependency of the microprocessor complex on Heme and makes a strong case that Heme does not controls the dimerization of DGCR8, as previously reported. The presented experiments are well designed with good controls, the quality of the figures is high and the paper is well written.

We thank the reviewer for the kind compliments. We have tried our best to test our hypotheses thoroughly by using many miRs and controls. We have further improved the manuscript taking the reviewers' suggestions and we hope it is ready for publication.

The only experiment, which I think should be improved before publication is the demonstration of the mechanism being active in transfected cells, which is based on detection of the levels of mature miRNA expressed from a transfected construct (figure 2b). I find that it is important to show that the cleavage positions of the Heme dependent miRNAs in cells are indeed dependent on Heme-loaded DGCR8. Preferably, this could be by performing small-RNA sequencing on transfected cells. Alternatively, the faulty processing of Heme-dependent miRNAs in cells could be demonstrated by detecting the miRNAs and pre-miRNA with northern.

We have attempted to detect the alternative cut site product using qPCR, splinted ligation, and Northern blotting, from the same samples where we quantify the mature miRs (Fig 2), but we have been unable to do so thus far. We postulate that the alternative product is highly unstable and prone to degradation in cells, thereby reducing the pool of available pri-miRNAs. As shown in the original draft, we consistently observe that the C352S mutation affects heme-dependent microRNA levels more dramatically in intact cells, even when expressed in tandem with the heme-independent ones. In our revised manuscript we also include splinted ligation assay data (Fig. 2d-g) that corroborate with the qPCR results. We used this assay because it was shown to be more sensitive than Northern to detect small RNAs (Maroney et al., 2008, Nat Protoc). For example, the pre-miRNA levels were also too low for detection by Northern despite numerous attempts. Finally, as the reviewer suggested, we also show that the heme-binding mutation affects microRNA levels broadly, by using small-RNA sequencing, in Supplementary Figure 2f (Also see major point 1 below).

Major points:

1. In my opinion, the manuscript would be significantly improved if the in-cell validation experiment presented in figure 2b-d were supplemented with a sequencing based analysis of the small RNA content of the cell. This would make it possible to investigate many more miRNAs than the ones presented in the manuscript and could allow the aberrant cleavage events for the Heme dependent pri-miRNAs to be directly identified, which would

demonstrate the in vivo relevance of the convincing in-vitro findings of the manuscript. Moreover, identification Heme dependent processing of endogenous pri-miRNAs is important to determine how wide-spread the phenomenon of Heme dependent processing is.

We now include data from a small-RNA sequencing experiment, to determine the effects of a heme binding mutation (C352S), in Supplementary Figure 2f. With a single point mutation that impairs heme binding activity of DGCR8, we observe that many endogenous miR levels change to varying degrees. Due to potential feedback loops in an intact cell, it is not easy relate the in vivo results directly to in vitro data. Nevertheless, our data do show that interfering with heme binding can have a widespread impact on miR profiles in cells.

As we mention above, we were unable to detect any aberrant cleavage events from this data, likely due to low abundance.

2. Statistical analysis of the data presented in figure 2b should be included.

We have updated the figure and include statistical analyses using 3 biological replicates. In addition, the results from individual transfections are shown separately in Supplementary Fig. 2c-d. To assist the reader, we now show the relative abundance normalized to the heme-independent control miRNA.

3. In the closing remark of the manuscript the authors speculate that the Heme dependence of some miRNA may act as a means for regulation by changes in oxidation state or gas levels. This is indeed a highly interesting possibility, which would be relatively easy for the authors to investigate with their well established in vitro setup.

Although we have established a clean in vitro setup, the large polypeptides that we have to use are highly sensitive. We observe that adding a strong reductant such as dithionite that is required to change the oxidation state of iron leads to precipitation of the proteins. As the reviewer mentioned, what we discovered as a molecular role of heme in Microprocessor function makes these extra regulatory pathways more interesting. Yet, what we would have to do to investigate the matter thoroughly requires many experiments and is beyond the scope of this study.

Minor comments:

1. The mechanism by which DGCR8 are able to detect the loop structure of the pri-MiRNAs is interesting. A more careful analysis of the loop requirements for DGCR8-Heme binding could possibly provide more insights. The SHAPE-probing clearly shows that the presence of DGCR8-Heme make specific positions in the pri-miRNA loops hyper reactive. The authors suggest that “remodeling of the RNA backbone due to direct interactions with protein” is causing the hyperreactivity. However, a larger remodeling of the loop would likely result in more positions changing reactivity than what is observed here. Hyper reactivity to SHAPE reagents is not observed that frequently and is typically caused either by the position being locked in a rare reactive backbone conformations or by base catalysis of the reaction (McGinnis et al, JACS 2012). I therefore think that it would be relevant to include some more discussion of how the Heme loaded DGCR8 could cause the hyperreactivity.

We agree with both of the reviewers that the SHAPE hotspot is indicative of a binding event, but that the actual interacting region is almost certainly significantly larger. Our results are reminiscent of SHAPE experiments showing hyperreactivity patterns in certain RNA protein complexes (McGinnis et al., 2012, and Smola et al., 2015), and obtaining structural information on the DGCR8/pri-miRNA interaction would yield very interesting insight into this observation. We have revised the text (p. 12) with elaboration on what may be causing this hyperreactivity and how it likely represents a significantly larger RNA/protein footprint.

2. The mature miRNA 5' end is defined by either Drosha or Dicer. It would be interesting, if the Heme dependence of correct Drosha processing correlates with the need for accurate Drosha definition of the seed sequence of different miRNAs. Is this the case?

We appreciate the reviewer's suggestion. We have checked the trend with the miRNAs that we have established as heme-dependent vs. heme-independent, but we do not observe any correlation between heme-dependency and which strand is selected as the mature miRNA. Dicer has been proposed to identify its cut site by measuring the distance from the Drosha cut site. Therefore, compromised Drosha accuracy is likely to affect most miRNAs that require canonical processing, regardless of which strand is selected as the mature product.

3. P8, line 15-16: please elaborate on how more degradation could lead to Heme dependent dimerization.

In our experience, absence of heme can lead to more degradation of DGCR8. We postulate that shortening of the polypeptide in the absence of heme might have been interpreted as heme-free "monomer". Our biophysical studies using multiple methods presented in this manuscript establish that DGCR8 dimerizes regardless of the presence of heme or the presence of Drosha. We have revised the text (p.9 bottom) to clarify this point.

4. In figure 1: The use of green/red coloring to designate Heme vs. non-Heme is not optimal and makes it difficult for the red/green color insecure individuals to interpret the figure.

Thank you for bringing this to our attention. We have changed all the figures to address this issue.

In addition, panel b could go to supplementary, whereas it would make sense to show the data now presented in in supplementary figure 1c in the main figure, as it is the most direct evidence of the MP flipping around.

We have incorporated the reviewer's suggestion and rearranged Figure 1 and Supplementary Figure 1.

5. P7, line 2, p8 line 21: "correct Drosha" is used, but "guide Drosha to the correct binding site" or similar would be more precise.

The text has been revised to clarify this section.

6. Supplementary Fig. 5a should include the non-SHAPE probed controls and the loop basepairing should only be indicated if the probing pattern support it. Information about the ladder used for these blots should be included.

The controls with no added SHAPE reagent (- BzCN) for pri-let-7d are included in Main Figure 5a. Due to the reviewer's comment we also revised the label to ensure clarity. For most of the Supplementary figure 5, we do not show the controls with no added SHAPE reagent for space reasons. As shown in the lanes without any added protein, the background signal we observe--even in the presence of the SHAPE reagent--is significantly less than what is observed in the lanes with heme-bound proteins. Since our study is merely focused on these obvious hotspots rather than using SHAPE for traditional purposes, we only show one example of the background in Figure 5a, while including the higher background lane (without protein but with SHAPE reagent). For similar reasons, all our loop diagrams are from mfold predictions. Our gel-based method may not be quantitative enough for a robust SHAPE-based RNA structure predictions. We understand the reviewer's concern that the including the structure next to the gel might mislead the reader to assume that it is empirically derived. Thus, we have revised the legend to clarify that the loop diagrams are there to help the reader locate

the hotspot in relation to the mature sequence. Finally, more information about the ladder has been added to the methods section describing our SHAPE experiments.

7. The SHAPE probing pattern of the point mutants investigated in Supplementary Fig. 5j-m would be an interesting addition to the paper.

We thank the reviewer for the interesting suggestion. Thus we followed up on the point mutation in let-7d as it had a larger effect on RNA binding affinity measured by EMSA, as shown in Supplementary figure 6c-d (old Supp Fig 5j-k). The new SHAPE analysis is now presented in Supplementary Fig. 6g. Interestingly, decrease in the hotspot intensity correlates with the loss of affinity for RNA. We had also revised the main text accordingly.

8. P12 line 16-18: this statement is too confident, given that only a single precursor was investigated.

The reviewer makes an important point that we have only quantified the contribution of loop structure vs. sequence using mir-9. Thus, we have revised the text (p. 15 lines 5-8) to acknowledge that both sequence and structure are likely to contribute to specificity.

9. P17 line 4: describe more carefully the method used for Q-PCR normalisation.

The qPCR figure has been updated and the methods section has been revised to clarify how we performed the normalization.

Reviewers' Comments:

Reviewer #1:

Remarks to the Author:

The authors have adequately addressed my comments or concerns. I recommend publication at this time.

Reviewer #2:

Remarks to the Author:

Review of "Heme enables proper positioning of Drosha and DGCR8 on primary microRNAs" manuscript revision.

The authors have made a number of changes to the manuscript and these changes largely address the points that was raised in my initial review. The changes have further improved the manuscript, which I think is an important addition to the field.

However, as acknowledged by the authors, the result of the sequencing experiment that now has been included is somewhat inconclusive and does not provide further support the heme effect in vivo. While the splint-ligation and Q-PCR results for miR-21 and miR-9-1 are convincingly showing an effect of expression of the mature form, I still think that the manuscript would be significantly improved, if the effect could be demonstrated on a more global scale and especially if the heme dependent processing could be shown.

The methods for the included sequencing experiment are not very detailed, making it difficult to know what to expect from this experiment. It is unclear if size selection was performed on the library and the details of mapping and expression analysis (replicates? and statistical analysis?) are not provided. Mir-21 and -18 seem to support the heme dependent processing, whereas miR-16 surprisingly shows the same effect. I agree that there could be downstream effects and feedback loops potentially explaining the results obtained here, but from what is now presented it is impossible to judge.

I therefore recommend that the sequencing experiment is performed again before the manuscript is published, but focusing on the precursor size range, which should make it possible to detect the changed precursor processing, even if these products are present at very low levels. miR-21, which is one of the heme dependent miRNAs according to fig 2, is expressed in Hek293 at a quite high level.

Responses to the reviewers' comments

We thank the reviewers for taking the time to review our paper. We appreciate that both of the reviewers share our enthusiasm for the work presented in the manuscript, as it elucidates a heme-dependent mechanism that helps us shape a molecular model for how Microprocessor works. We are also thankful for all the helpful suggestions, as following them has enabled us to improve our manuscript further.

Reviewers' comments:

Reviewer #1 (Remarks to the Author):

The authors have adequately addressed my comments or concerns. I recommend publication at this time.

We thank the reviewer again for helpful suggestions to improve the manuscript.

Reviewer #2 (Remarks to the Author):

Review of "Heme enables proper positioning of Drosha and DGCR8 on primary microRNAs" manuscript revision.

The authors have made a number of changes to the manuscript and these changes largely address the points that was raised in my initial review. The changes have further improved the manuscript, which I think is an important addition to the field.

We thank the reviewer again for helpful suggestions to improve the manuscript.

However, as acknowledged by the authors, the result of the sequencing experiment that now has been included is somewhat inconclusive and does not provide further support the heme effect in vivo. While the splint-ligation and Q-PCR results for miR-21 and miR-9-1 are convincingly showing an effect of expression of the mature form, I still think that the manuscript would be significantly improved, if the effect could be demonstrated on a more global scale and especially if the heme dependent processing could be shown.

We agree with the reviewer that it is helpful to have more evidence to show that heme affects processing of primary transcripts in intact cells. Previously we inferred this by quantifying the mature miR levels derived from over-expressed, tandem pri-miRs. Recently, we succeeded in detecting pre-miR levels by using Northern blotting, using the same transiently transfected cells, although pre-miR detection is noisier than for mature miRs (new Fig. 2i-2l). In addition, we also quantified the amount of pri-miRs using quantitative PCR (new Fig. 2h). Thus, when the cells overexpress the designated tandem pri-miRs, we can measure the amounts of the substrates and the products of Drosha, allowing us to estimate the in vivo processing efficiency of a particular miR. As predicted from our model (Fig 6), our new results show that the pre-miR levels decrease for both pri-miRs, but heme-dependent miRs are affected more dramatically because fidelity is comprised in addition to lower affinity. In contrast, pri-miR levels for both classes of miRs stay similar, or even somewhat stabilized in the cells expressing mutant DGCR8. Thus, we now have evidence showing that heme affects the processing step from pri-miR to pre-miR in cells, in agreement with our in vitro studies. Together, our results show that heme enhances maturation of most miRs, but that "heme-dependent" pri-miRs are more dramatically affected by heme bound to Microprocessor. We have edited the main text to include the discussion of these results. We thank the reviewer for strongly encouraging us to obtain more data to show that heme affects processing of pri-miRs in cells, as our new data strengthens our model.

The methods for the included sequencing experiment are not very detailed, making it difficult to know what to expect from this experiment. It is unclear if size selection was performed on the library and the details of

mapping and expression analysis (replicates? and statistical analysis?) are not provided. Mir-21 and -18 seem to support the heme dependent processing, whereas miR-16 surprisingly shows the same effect. I agree that there could be downstream effects and feedback loops potentially explaining the results obtained here, but from what is now presented it is impossible to judge. I therefore recommend that the sequencing experiment is performed again before the manuscript is published, but focusing on the precursor size range, which should make it possible to detect the changed precursor processing, even if these products are present at very low levels. miR-21, which is one of the heme dependent miRNAs according to fig 2, is expressed in Hek293 at a quite high level.

We apologize that we did not include enough details on the sequencing experiment in the previous version of the manuscript. We had performed size selection to include both the mature miRs and the pre-miRs, and we had 3 biological replicates each for WT and the heme-binding mutant DGCR8. To prevent noise from long-lived miRs, we also performed two consecutive transfections with the DGCR8 constructs before harvesting the RNA sample to sequence. In our previous version of the manuscript, we only presented the data for mature miRs to illustrate how the heme-binding mutation affects steady state levels of many mature miRs. As the reviewer suggested, we hoped that the steady state pre-miR levels would be readily detectable and change with heme-bound state of Microprocessor. However, the data is rather inconclusive. First, quantitation of pre-miR levels by sequencing is not as robust as mature miRs, perhaps due to bias during ligation and also because certain pre-miRs are especially difficult to reverse-transcribe. For example, for miR-21 where we see nearly 50k reads for the mature form, we observe only 1 read for the pre-miR. We found that only ten pre-miRs had more than 5 total reads in the wild type sample (Reviewer Fig 1). Due to the small sample size, it is difficult to perform a statistical analysis on the pre-miR levels. Moreover, we do not observe a correlation between mature and pre-miR levels (Reviewer Fig 2). We do not think deeper sequencing depth or narrower size selection would help since similar challenges have been observed in such sequencing experiments (Li, 2013).

In addition to the technical challenges in quantifying pre-miRs by sequencing, we also believe that a systems approach is necessary to thoroughly assess the global effect of heme on pri-miR processing. The level of a pre-miR can change with heme due to several factors: 1) erroneous, dead-end, processing of pri-miR; 2) changes in the overall affinity and enzymatic efficiency (rate) of Microprocessor for that pri-miRNA; 3) changes in the abundance of concurrently expressed pri-miRs; and/or 4) heme-dependence of concurrently expressed pri-miRs. Moreover, a heme-dependent miR may also affect processing rates of other miRs in the same cluster. Furthermore, it is also unclear how the half-life of a pre-miR changes with changing levels or populations of other pre-miRs. Finally, many proteins can regulate expression and processing of pri-miRs and their levels can change due to changing levels of miRs. In Fig 2, we were able to bypass such complexities and performed a limited quantitative analysis by using two vastly overexpressed pri-miRs. But given the varying effects of heme on each pri-miR, to perform a global analysis of endogenous pri-miR processing thoroughly would require a systems approach including many sequencing experiments (to measure pri-, pre-, and mature), time points, and perturbation methods, which is beyond the scope of this study. For similar reasons, the mature miR sequencing data we presented in the previous version may be oversimplified and may confuse the reader. As the reviewer concurs with us that the sequencing data does not readily extend our findings, we have removed the data from the manuscript. We agree with the reviewer that global analyses of heme effects is an important next step given our findings on how heme regulates miR biogenesis at the molecular level. We hope that publication of our manuscript will help stimulate such investigations in the future.

Reference:

Li, N., You, X., Chen, T., Mackowiak, S. D., Friedländer, M. R., Weigt, M., et al. (2013). Global profiling of miRNAs and the hairpin precursors: insights into miRNA processing and novel miRNA discovery. *Nucleic Acids Research*, 41(6), 3619–3634. <http://doi.org/10.1093/nar/gkt072>

Figures for the reviewers only:

Figure 1. Relative abundances of pre-miRs after overexpression of either wild type or C352S DGCR8.

Figure 2. Scatter plot illustrating number of reads of pre-miR and corresponding mature miR

Reviewers' Comments:

Reviewer #2:

Remarks to the Author:

The new data included in figure 2 has improved the revised manuscript and support the authors conclusions, while still not demonstrating that the heme dependent processing (as outlined in the model, fig 6) occurs in cells. The in vitro data is very strong and support the model, but things could be more complex in vivo. I still think that a global dedicated sequencing based analysis of precursor sized RNA would be possible and be an important addition to the paper, but I also buy the point that this could be outside the scope of this paper. I support the publication of the manuscript, if the authors could add a short section to the discussion, acknowledging that the the model, while fitting the available data, still needs to be validated in vivo.

Responses to the reviewers' comments

Reviewers' comments:

The new data included in figure 2 has improved the revised manuscript and support the authors conclusions, while still not demonstrating that the heme dependent processing (as outlined in the model, fig 6) occurs in cells. The in vitro data is very strong and support the model, but things could be more complex in vivo. I still think that a global dedicated sequencing based analysis of precursor sized RNA would be possible and be an important addition to the paper, but I also buy the point that this could be outside the scope of this paper. I support the publication of the manuscript, if the authors could add a short section to the discussion, acknowledging that the the model, while fitting the available data, still needs to be validated in vivo..

As the reviewer suggested, we have edited the last paragraph in the Discussion section to clearly state that more in vivo studies are required to show how heme might regulate microRNA processing in vivo. We also add that there may be additional layers to the regulatory mechanism, such as crosstalk with other pathways and through various forms of heme. We are happy that the reviewer finds our manuscript acceptable for publication, and we thank the reviewer again for all the suggestions, as they helped us to improve the manuscript.